# Exact Multi-objective Path Finding with Negative Weights and Negative Cycles

**Primary Keywords:** *None*

## Abstract

The point-to-point Multi-objective Shortest Path (MOSP) problem is a classic yet challenging task that involves finding all Pareto-optimal paths between two points in a graph with multiple edge costs. Recent studies have shown that employing A* search can lead to state-of-the-art performance in solving point-to-point MOSP instances with non-negative costs. In this paper, we propose a novel A*-based multi-objective search framework that not only handles graphs with negative costs and even negative cycles but also incorporates multiple speed-up techniques to enhance the efficiency of exhaustive search with A*. Through extensive experiments on large realistic test cases, our algorithm demonstrates remarkable success in solving difficult MOSP instances, outperforming the state of the art by up to an order of magnitude.

## Introduction

The point-to-point Multi-objective Shortest Path Problem (MOSP) is a classic network optimisation problem that involves finding all Pareto-optimal paths between a pair of (*Origin*, *Destination*) locations in graphs with multiple link attributes. The problem has a wide range of real-world applications in diverse areas such as transportation planning, telecommunication networks, and robotics. MOSP can be modelled to plan paths that are optimal in terms of fuel consumption, distance, and arrival time in maritime transportation (Wang, Mao, and Eriksson 2019), to select emergency routes for major chemical accidents (Xu, mei Gai, and Salhi 2021), or to simultaneously minimise difficulty, risk, and elevation of planned paths for mobile robots in harsh situations (Jeddisaravi, Alitappeh, and Guimarães 2016).

MOSP and its bi-objective variant BOSP (Bi-objective Shortest Path) are well-studied topics in both network optimisation and AI literature, and have attracted growing interest in recent years. Salzman et al. present an overview of some of the existing MOSP algorithms and their key features, highlighting the significance of heuristic-guided search, in particular A* (Hart, Nilsson, and Raphael 1968), in reducing overall computation time of MOSP. BOSP can be considered as the most basic variant of MOSP. Currently, there are several specialised algorithms designed to tackle BOSP on a large scale, such as Bi-objective search with A* (Ulloa et al. 2020; Ahmadi et al. 2021a). Among the recent attempts at optimally solving MOSP for more than two ob-

jectives, three novel exact solutions, namely EMOA* (Ren et al. 2022), TMDA (Maristany de las Casas et al. 2023), and LTMOA* (Hernández et al. 2023), have been successful in efficiently utilising best-first search to address the problem. EMOA* follows the search strategy of NAMOA*$_{dr}$ of Pulido, Mandow, and Pérez-de-la-Cruz (2015a) but employs balanced binary search trees to store non-dominated partial paths expanded during the search. The recent TMDA algorithm adapts one-to-all MDA of de las Casas, Sedeño-Noda, and Borndörfer (2021) to a point-to-point MOSP solution using heuristic-guided search. Despite NAMOA*$_{dr}$ where the search priority queue contains all unexplored paths, the multi-objective search of TMDA follows a Dijkstra-like queueing approach (Dijkstra 1959) and stores at most one (best) path per graph node into the priority queue. LTMOA* is another A*-based algorithm that performs linear-time dominance check prior to expanding partial paths. The dominance check is necessary to insures the (partial) path is not dominated by any previously expanded path in all objectives, thus reducing the search effort by avoiding unnecessary expansions. Based on the results, it is evident that all EMOA*, TMDA, and LTMOA* outperform NAMOA*$_{dr}$, with LTMOA* demonstrating superior performance compared to EMOA*. However, a direct comparison between LTMOA* and TMDA is currently unavailable.

**MOSP with negative weights and negative cycles**: Many real-world applications of MOSP need to be modelled with graphs containing negative edge weights. Energy requirement, for example, can be observed in both positive (consumption) and negative (generation) forms. Unfortunately, none of the aforementioned point-to-point MOSP solutions are capable of handling such graphs. While there are a few existing solutions to MOSP with negative weights, it has remained a relatively underexplored topic. The one-to-all MDA algorithm of de las Casas, Sedeño-Noda, and Borndörfer (2021), build on the basis of Martin's Principle of Optimality (Martins 1984), can solve MOSP with negative weights. MDA can only solve bounded MOSP instances. Thus, the graph must be free of negative cycles (Sastry, Janakiraman, and Mohideen 2003). Note that one-to-all MOSP exhibits a larger search space than the point-to-point variant, demanding far larger computation time. The path-ranking method of Sastry, Janakiraman, and Mohideen (2005) and the label setting approach of Kurbanov, Cuchý,

and Vokrínek (2022) are two other MOSP approaches that can deal with negative weights. These solutions are not exact as they do not compute all Pareto-optimal solutions.

This paper introduces NWMOA$^\star$, a point-to-point Multi-Objective A* search framework that can deal with both Negative Weights and negative cycles. NWMOA$^\star$ adapts the best-first search strategy of A* to problems with negative attributes, and introduces novel speedup techniques to further enhance the search performance, including time-efficient dominance test and queueing strategies. The result of our extensive experiments on a new set of large realistic instances show the success of NWMOA$^\star$ in solving large MOSP instances faster than all existing methods.

## Notation and Problem Formulation

Consider a MOSP problem provided as directed graph $G = (S, E)$ with a finite set of states $S$ and a set of edges $E \subseteq S \times S$. Every edge $e \in E$ of the graph has $k \in \mathbb{N}$ attributes that can be accessed via the cost function $\mathbf{cost} : E \to \mathbb{R}^k$ and we have $\mathbf{cost} = (cost_1, cost_2, \ldots, cost_k)$ as a form of vector. A path is a sequence of states $s_i \in S$ with $i \in \{1, \ldots, n\}$. The $\mathbf{cost}$ vector of path $\pi = \{u_1, u_2, u_3, \ldots, u_n\}$ is then the sum of corresponding attributes on all the edges constituting the path, namely $\mathbf{cost}(\pi) = \sum_{i=1}^{n-1} \mathbf{cost}(u_i, u_{i+1})$. Since costs can be negative values, we say path $\pi$ forms an elementary negative cycle on $cost_i$ if i) $cost_i(\pi) < 0$; ii) $u_i \neq u_j$ for any $i, j \in \{1 \ldots n-1, i \neq j\}$; iii) $u_1 = u_n$. The point-to-point MOSP aims to find a set of cost-unique Pareto-optimal paths between a given pair of $start \in S$ and $goal \in S$, a set in which every individual solution offers a path that minimises the multi-criteria problem in all dimensions.

Following the conventional notation in the heuristic search literature, we define our search objects to be *nodes* (equivalent to partial paths). A node $x$ is a tuple that contains the main information of the partial path to $s(x)$, where $s(x)$ is a function returning the state associated with $x$. Node $x$ traditionally stores a cost vector $\mathbf{g}(x)$ which measures the $\mathbf{cost}$ of a concrete path from the $start$ state to state $s(x)$. In addition, node $x$ is associated with the cost vector $\mathbf{f}(x)$, which estimates the $\mathbf{cost}$ of a complete path from $start$ to $goal$ via $s(x)$; and also a reference $parent(x)$ which indicates the parent node of $x$. Further, the operator $\text{Tr}(\mathbf{v})$ truncates the $cost_1$ of the cost vector $\mathbf{v}$. For example, $(g_2(x), \ldots, g_k(x))$ is the truncated vector of $\mathbf{g}(x)$.

We consider all operations of the cost vectors to be done element-wise. For example, we define $\mathbf{g}(x) + \mathbf{g}(y)$ as $(g_1(x) + g_1(y), \ldots, g_k(x) + g_k(y))$. We use $\preceq$ or $\leq_{lex}$ symbols in direct comparisons of cost vectors, e.g. $\mathbf{g}(x) \preceq \mathbf{g}(y)$ denotes $g_i(x) \leq g_i(y)$ for all $i \in \{1, \ldots, k\}$ and $\mathbf{g}(x) \leq_{lex} \mathbf{g}(y)$ means the cost vector $\mathbf{g}(x)$ is lexicographically smaller than or equal to $\mathbf{g}(y)$. Analogously, we use $\not\prec$ or $\not\leq_{lex}$ symbols if the relations cannot be satisfied.

**Definition** For the node pair $(x, y)$, we say $y$ is (weakly) dominated by $x$ if we have $\mathbf{g}(x) \preceq \mathbf{g}(y)$ or $\mathbf{f}(x) \preceq \mathbf{f}(y)$.

The main search in A*-based solution methods is guided by $start$-$goal$ cost estimates or $\mathbf{f}$-values, which are traditionally established based on a consistent and admissible heuristic function $\mathbf{h} : S \to \mathbb{R}^k$ (Hart, Nilsson, and Raphael 1968). In other words, for every search node $x$, we have $\mathbf{f}(x) = \mathbf{g}(x) + \mathbf{h}(s(x))$ where $\mathbf{h}(s(x))$ estimates lower bounds on the $\mathbf{cost}$ of paths from state $s(x)$ to the $goal$ state. One common method of computing a well-informed $h_i$ function is solving a one-to-all single-objective shortest path problem on $cost_i$ from the $goal$ state on the graph $G$ with all links reversed, building a *perfect heuristic* function. A perfect heuristic is consistent and admissible.

**Definition** $h_i : S \to \mathbb{R}$ is admissible iff $h_i(u) \leq cost_i(\pi^*)$ for every $u \in S$ where $\pi^*$ is the optimal path on $cost_p$ from state $u$ to the $goal$ state. It is also consistent if we have $h_i(u) \leq cost_i(u, v) + h_i(v)$ for every edge $(u, v) \in E$.

## Multi-objective Search with A*

The multi-objective search with A* involves in general two types of strategies: expand and prune. It performs a systematic search by *expanding* nodes in best-first order. That is, the search is led by a partial path that shows the lowest $\mathbf{cost}$ estimate or $\mathbf{f}$-value. Each iteration involves three main steps: i) Extraction: remove one (lexicographically) least-$\mathbf{cost}$ node from a priority queue, known as $Open$ list. ii) Dominance check: ensure the extracted node is not dominated by some previous expansions; iii) Expansion: generate new (non-dominated) descendant nodes and store them in $Open$ for further expansion.

To commence the search, we initialise $Open$ with a node associated with $start$, $\mathbf{g} = \mathbf{0}$ and $\mathbf{f} = \mathbf{h}$. $Open$ always contains generated (but not expanded) nodes. For the purpose of further expansion, $Open$ reorders its nodes according to their $\mathbf{f}$-value such that the lexicographically least-$\mathbf{cost}$ node is always at the front of the list. More accurately, given the cost vector $(f_1, f_2, \ldots, f_k)$, $Open$ first orders nodes based on their $f_1$-value, and then the truncated vector $(f_2, \ldots, f_k)$ if it finds two (or more) of the nodes showing the same $f_1$-value. The latter operation is called *tie-breaking*. Once a least-$\mathbf{cost}$ node is extracted from $Open$, it undergoes a rigorous dominance test to ensure the expansion of the node can lead to a promising solution. This dominance rule usually involves checking the $\mathbf{cost}$ of the extracted node against that of the previously expanded nodes of the state, as well nodes expanded with the $goal$ state (established solutions). The same strategy can be applied once a new descendant node is generated (during expansion). It is always safe to prune dominated nodes, essentially because their expansion will never lead to an optimal solution. Nodes associated with the $goal$ state represent solution paths and can be stored in a solution set. Since solution paths must be acyclic, A* does not need to expand such solution nodes. Finally, multi-objective A* terminates once there is no node in $Open$ to explore. With this introduction, we now describe our NWMOA$^\star$.

## Multi-objective Search with Negative Weights and Negative Cycles

Many real-world MOSP problems deal with attributes that are negative in nature, or attributes that may exhibit negative values in specific circumstances, such as energy recuperation in electric vehicles. There might also be cases where the graph contains negative cycles on (one or even all) cost com-

ponents. Our NWMOA* performs multi-objective search on the basis of A* and can deal with negative weights and negative cycles. We start with discussing the latter case.

**Negative cycles:** Although it is widely accepted in the MOSP literature that the existence of negative cycles makes MOSP inevitably unbounded, we now elaborate this is not always the case in the point-to-point variant. Consider the sample graph of Figure 1 with three attributes shown on the edges. The graph contains negative weights, as well as negative cycle (through the third cost component of the links $u_4 \rightsquigarrow u_5 \rightsquigarrow u_6 \rightsquigarrow u_4$ shown in red). We aim to find optimal paths from $u_s$ to $u_g$. This MOSP instance is bounded, as there are three optimal paths, none of them visiting the vertices of the negative cycle. Given this important observation, we can conclude that negative cycles are not problematic for a point-to-point MOSP instance as long as they do not appear on any $start\text{-}goal$ path. Thus, we distinguish two cases:

1. There is negative cycle on any arbitrary path from $start$ to $goal$: The problem becomes unbounded, as there will be at least one dimension on which we can take the negative cycle to further reduce cost of the path indefinitely.

2. No path can be found from $start$ to $goal$ via a negative cycle: The problem is bounded, as this essentially implies that there is either no negative cycle present, or if there is one, it cannot be reached from $start$ or reach $goal$.

As described above, to determine whether the problem is bounded or not, we need to perform a simple (polynomial-time) reachability test: Let $S'$ be the set of states that are reachable from $start$ and can reach $goal$. The problem is unbounded if we can find a negative cycle on $G' = (S', E')$ with $E' \subseteq S' \times S'$. In the sample graph of Figure 1, the three states consisting the only negative cycle of the graph can reach $u_g$ (the $goal$ state), but are not reachable from $u_s$ (the $start$ state). Hence, we can assure that no negative cycle can appear on any optimal path, and thus the multi-objective search can be conducted safely. To accommodate this crucial preliminary test, we describe our NWMOA* in two levels.

### NWMOA*'s High-Level Description:

Algorithm 1 provides a high-level description of NWMOA*, presenting it as a merged procedure for establishing heuristic functions and detecting negative cycles. It first initialises **h**-value of all states with a vector of $k$ large values, followed by eliminating all states not reachable from $start$ (e.g., using breath-first search) to form a reduced graph. The algorithm then conducts for each $i \in \{1, \ldots, k\}$ a backward one-to-all single-objective search to compute $cost_i$-optimal path from $goal$. This can be as simple as $k$ runs of the Bellman-Ford algorithm (Bellman 1958; Ford Jr 1956), or the Dijkstra's algorithm with re-expansions allowed (Johnson 1973). There will be negative cycle on a $start\text{-} goal$ path if the length (number of edges) of the $cost_i$-optimal path to any state of the reduced graph grows to be larger than or equal to $|S|$. The condition $h_i(u) = -\infty$ at line 6 of Algorithm 1 denotes the situation where we can identify unbounded MOSP instances through the growing length of optimal paths. Otherwise, the problem is bounded and we can safely proceed with the NWMOA*'s lower-level search (via line 8).

---

**Algorithm 1:** NWMOA* High Level

**Input:** A MOSP Problem ($G$, $start$, $goal$, $k$)
**Output:** A cost-unique Pareto-optimal solution set

1   $\mathbf{h}(u) \leftarrow \infty \ \forall u \in S$
2   $S \leftarrow$ Remove from $S$ states not reachable form $start$
3   **for** $i \in \{1 \ldots k\}$ **do**
4     **foreach** $u \in S$ **do**
5       $h_i(u) \leftarrow cost_i$-optimal path from $u$ to $goal$
6       **if** $h_i(u) = -\infty$ **then**
7         **return** $\emptyset$

8   $Sols \leftarrow$ Multi-objective Search on ($G$, $\mathbf{h}$, $start$, $goal$)
9   **return** $Sols$

---

### Multi-objective Search of NWMOA*:

Our new search framework differs from existing methods in three key aspects:

i) Nodes in NWMOA* are processed in order of their $f_1$-value, rather than any lexicographical ordering. This will reduce the cost of priority queue operations, and enable the use of simpler yet efficient queueing data structures.

ii) The search is equipped with a novel constant-time dominance check. This will help reduce the overall dominance check attempts, and nodes generated during the search.

iii) Unlike LTMOA*, where truncated vectors are stored in no specific order, NWMOA* stores truncated vectors in lexicographical order, reducing dominance checks per iteration.

Algorithm 2 shows the pseudo-code of the multi-objective search of NWMOA*, with the new features highlighted in blue. The algorithm starts with initialising the priority queue $Open$, and the solution set $Sols$. It then sets up for every state $u \in S$ a list $\mathrm{G}_{\mathrm{cl}}^{\mathrm{Tr}}(u)$, responsible for storing the (non-dominated) truncated cost-vector of previous (closed) expansions with state $u$. To reduce the number of costly dominance checks, NWMOA* keeps track of the most recent expansion of $u$ via a cost vector called $\mathbf{g}_{\mathrm{last}}^{\mathrm{Tr}}(u)$. This cost vector is initialised with large cost values (line 3) to allow for capturing the first expansion. NWMOA* then initialises a node with the $start$ state and insert it into the priority queue.

Each iteration of the algorithm starts at line 7. Let $Open$ be a non-empty queue. NWMOA* extracts in each iteration of the algorithm a node $x$ with the smallest $f_1$-value (line 8).

**Quick dominance check:** Most recent expansions are generally more informed and can be seen as strong candidates for dominance check. Node $x$ is quickly checked for dominance against the last expanded node of two states: $goal$ and the state associated with $x$, i.e., $s(x)$ (line 9). As we will show in the next section, NWMOA* processes nodes in non-decreasing order of their $f_1$-value. Because the first dimension is expanded in sorted $f_1$ order, later expansions are already dominated by previous $f_1$-values. So, the dominance test can be done by just comparing the truncated cost vector of $x$, i.e., $\mathrm{Tr}(\mathbf{f}(x))$, with that of the two candidates.

**Dominance check with IsDominated:** If the extracted node cannot be quickly dominated, NWMOA* takes $\mathrm{Tr}(\mathbf{g}(x))$ and $\mathrm{Tr}(\mathbf{f}(x))$ to conduct a rigorous dominance check by comparing $x$ against (potentially all) truncated vec-

**Algorithm 2:** Multi-objective Search of NWMOA$^\star$

---

**Input:** A MOSP Problem $(G, \mathbf{h}, start, goal)$
**Output:** A cost-unique Pareto-optimal solution set

1   $Open \leftarrow \emptyset$ , $Sols \leftarrow \emptyset$
2   $G_{cl}^{Tr}(u) \leftarrow \emptyset$ $\forall u \in S$
3   $\mathbf{g}_{last}^{Tr}(u) \leftarrow \infty$ $\forall u \in S$
4   $x \leftarrow$ new node with $s(x) = start$
5   $\mathbf{g}(x) \leftarrow \mathbf{0}$ , $\mathbf{f}(x) \leftarrow \mathbf{h}(start)$ , $parent(x) \leftarrow null$
6   Add $x$ to $Open$
7   **while** $Open \neq \emptyset$ **do**
8      Extract from $Open$ node $x$ with the smallest $f_1$-value
9      **if** $\mathbf{g}_{last}^{Tr}(s(x)) \preceq \mathrm{Tr}(\mathbf{g}(x))$ **or** $\mathbf{g}_{last}^{Tr}(goal) \preceq \mathrm{Tr}(\mathbf{f}(x))$ **then continue**
10      **if** $\texttt{IsDominated}(\mathrm{Tr}(\mathbf{g}(x)), G_{cl}^{Tr}(s(x)))$ **or** $\texttt{IsDominated}(\mathrm{Tr}(\mathbf{f}(x)), G_{cl}^{Tr}(goal))$ **then**
11         **continue**
12      $\texttt{Consolidate}(\mathrm{Tr}(\mathbf{g}(x)), G_{cl}^{Tr})$
13      $\mathbf{g}_{last}^{Tr}(s(x)) \leftarrow \mathrm{Tr}(\mathbf{g}(x))$
14      **if** $s(x) = goal$ **then**
15         $i \leftarrow |Sols|$
16         **while** $i >= 1$ **do**
17            $z \leftarrow$ Node at index $i$ of $Sols$
18            **if** $f_1(x) \neq f_1(z)$ **then break**
19            **if** $x \preceq z$ **then** remove $z$ from $Sols$
20            $i \leftarrow (i - 1)$
21         Add $x$ to the end of $Sols$
22         **continue**
23      **foreach** $t \in Succ(s(x))$ **do**
24         $y \leftarrow$ new node with $s(y) = v$
25         $\mathbf{g}(y) \leftarrow \mathbf{g}(x) + \mathbf{cost}(s(x), t)$
26         $\mathbf{f}(y) \leftarrow \mathbf{g}(y) + \mathbf{h}(t)$
27         $parent(y) \leftarrow x$
28         **if** $\mathbf{g}_{last}^{Tr}(t) \preceq \mathrm{Tr}(\mathbf{g}(y))$ **or** $\mathbf{g}_{last}^{Tr}(goal) \preceq \mathrm{Tr}(\mathbf{f}(y))$ **then continue**
29         Add $y$ to $Open$
30   **return** $Sols$

---

**Algorithm 3:** $\texttt{IsDominated}$

---

**Input:** A cost vector $\mathbf{v}$ and a set of cost vectors $\mathbf{V}$
**Output:** $true$ or $false$

1   **for** $i \in \{1 \ldots |\mathbf{V}|\}$ **do**
2      $\mathbf{v}' \leftarrow$ Cost vector at index $i$ of $\mathbf{V}$
3      **if** $\mathbf{v}' \not\preceq_{lex} \mathbf{v}$ **then**
4         **return** $false$
5      **if** $\mathbf{v}' \preceq \mathbf{v}$ **then**
6         **return** $true$
7   **return** $false$

---

**Algorithm 4:** $\texttt{Consolidate}$

---

**Input:** A cost vector $\mathbf{v}$ and a set of cost vectors $\mathbf{V}$
**Output:** $\mathbf{V}$ updated

1   $i \leftarrow |\mathbf{V}|$
2   **while** $i \geq 1$ **do**
3      $\mathbf{v}' \leftarrow$ Cost vector at index $i$ of $\mathbf{V}$
4      **if** $\mathbf{v} \not\preceq_{lex} \mathbf{v}'$ **then**
5         **break**
6      **if** $\mathbf{v} \preceq \mathbf{v}'$ **then**
7         Remove $\mathbf{v}'$ from $\mathbf{V}$
8      $i \leftarrow (i - 1)$
9   Insert $\mathbf{v}$ to index $(i + 1)$ of $\mathbf{V}$
10   **return**

---

tors of previous expansions with both $s(x)$ and $goal$. Algorithm 3 shows the details of this procedure. $G_{cl}^{Tr}(s(x))$ stores in a lexicographical order all non-dominated truncated cost vectors derived from previous expansions with $s(x)$. Thus, the dominance test of Algorithm 3 does not need to traverse the entire $G_{cl}^{Tr}(s(x))$ list, and can terminate early when it discovers a candidate with a truncated cost vector not lexicographically smaller than that of $x$ (line 3 of Algorithm 3).

**Lexicographical ordering with** $\texttt{Consolidate}$: Let $x$ be a non-dominated node. Thus, its truncated cost vector, i.e., $\mathrm{Tr}(\mathbf{g}(x))$, must be stored in $G_{cl}^{Tr}(s(x))$ for the purpose of future dominance checks with $s(x)$. However, it is possible that $\mathrm{Tr}(\mathbf{g}(x))$ dominates some vectors of $G_{cl}^{Tr}(s(x))$. Truncated vectors lexicographically smaller than $\mathrm{Tr}(\mathbf{g}(x))$ cannot be dominated. Thus, $\texttt{Consolidate}$, described in Algorithm 4, iterates backward through $G_{cl}^{Tr}(s(x))$ to remove dominated vectors (line 7), and stops as soon as it finds $\mathrm{Tr}(\mathbf{g}(x))$ no longer lexicographically smaller than the candidate vector in $\mathrm{Tr}(\mathbf{g}(x))$. It then inserts $\mathrm{Tr}(\mathbf{g}(x))$ after the last attempted candidate (line 9), ensuring $G_{cl}^{Tr}(s(x))$ main-

tains its lexicographical order. A similar strategy is used in TMDA (Maristany de las Casas et al. 2023). Once $\mathrm{Tr}(\mathbf{g}(x))$ is captured, NWMOA$^\star$ stores a copy of it into $\mathbf{g}_{last}^{Tr}(s(x))$ as the most recent expansion (line 13 of Algorithm 2).

**Capturing solution paths:** Let $x$ represent a tentative solution path, i.e., we have $s(x) = goal$ (line 14 of Algorithm 2). If $Sols$ is empty, we simply capture $x$ as a tentative solution and add it to the solution set (line 21). Otherwise, since NWMOA$^\star$ does not explore nodes in a lexicographical order, it is possible for $x$ to dominate some previous solution nodes in $Sols$. NWMOA$^\star$ takes care of such situation via lines 15-20. To remove any dominated solutions from the list, NWMOA$^\star$ (linearly) iterates backward through $Sols$ and checks the new solution $x$ against those tied with $x$, i.e., those showing the same $f_1$-value. Solution nodes with $f_1$-value smaller than $f_1(x)$ cannot be dominated by $x$. Thus, a full traversal of $Sols$ may not be necessary (line 18).

**Expansion:** Let $x$ be a non-dominated node other than a solution node. Expansion of $x$ involves generating new descendant nodes through $s(x)$'s successors. Consistent with lazy dominance checks in recent MOSP solutions, NWMOA$^\star$ delays the (full) dominance check of descendant nodes until they are extracted from $Open$. Nonetheless, the quick dominance check against the most recent expansion of each successor state can still be applied (see line 28). Such quick pruning during expansion can reduce the queue size and consequently improve the search performance.

Finally, the algorithm returns $Sols$, as a cost-unique Pareto-optimal solution set to a bounded MOSP instance.

**Example:** We further elaborate on the key steps of

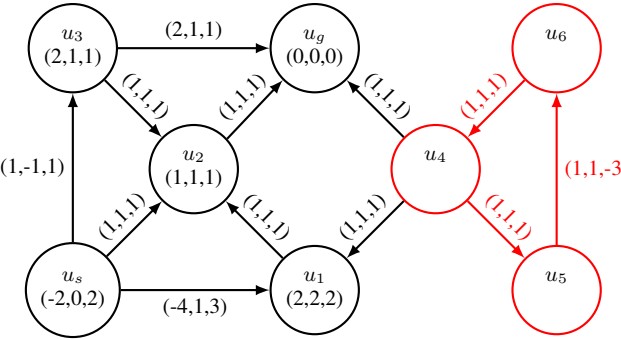

Figure 1: An example graph with three edge attributes and negative cycle. Triples inside the states denote **h**-value.

| It. | $Open : [\mathbf{f}(x), \mathbf{g}(x), s(x)]$ | $\mathrm{G}_{\mathrm{cl}}^{\mathrm{Tr}}$ | $Sols$ |
|---|---|---|---|
| 1 | $^*x_0$=[(-2,0,2), (0,0,0), $u_s$] | $\mathrm{G}_{\mathrm{cl}}^{\mathrm{Tr}}(u_s)$=[(0,0)] | |
| 2 | $^*x_1$=[(-2,3,5), (-4,1,3), $u_1$]
$x_2$=[(2,2,2), (1,1,1), $u_2$]
$x_3$=[(3,0,2), (1,-1,1), $u_3$] | $\mathrm{G}_{\mathrm{cl}}^{\mathrm{Tr}}(u_1)$=[(1,3)] | |
| 3 | $^*x_4$=[(-2,3,5), (-3,2,4), $u_2$]
$x_2$=[(2,2,2), (1,1,1), $u_2$]
$x_3$=[(3,0,2), (1,-1,1), $u_3$] | $\mathrm{G}_{\mathrm{cl}}^{\mathrm{Tr}}(u_2)$=[(2,4)] | |
| 4 | $^*x_5$=[(-2,3,5), (-2,3,5), $u_g$]
$x_2$=[(2,2,2), (1,1,1), $u_2$]
$x_3$=[(3,0,2), (1,-1,1), $u_3$] | $\mathrm{G}_{\mathrm{cl}}^{\mathrm{Tr}}(u_g)$=[(3,5)] | $x_5$ |
| 5 | $^*x_2$=[(2,2,2), (1,1,1), $u_2$]
$x_3$=[(3,0,2), (1,-1,1), $u_3$] | $\mathrm{G}_{\mathrm{cl}}^{\mathrm{Tr}}(u_2)$=[(1,1)] | $x_5$ |
| 6 | $^*x_6$=[(2,2,2), (2,2,2), $u_g$]
$x_3$=[(3,0,2), (1,-1,1), $u_3$] | $\mathrm{G}_{\mathrm{cl}}^{\mathrm{Tr}}(u_g)$=[(2,2)] | $x_{5,6}$ |
| 7 | $^*x_3$=[(3,0,2), (1,-1,1), $u_3$] | $\mathrm{G}_{\mathrm{cl}}^{\mathrm{Tr}}(u_3)$=[(-1,-1)] | $x_{5,6}$ |
| 8 | $^*x_7$=[(3,1,3), (2,0,2), $u_2$]
$x_8$=[(3,0,2), (3,0,2), $u_g$] | $\mathrm{G}_{\mathrm{cl}}^{\mathrm{Tr}}(u_2)$=[(0,2),(1,1)] | $x_{5,6}$ |
| 9 | $^*x_9$=[(3,1,3), (3,1,3), $u_g$]
$x_8$=[(3,0,2), (3,0,2), $u_g$] | $\mathrm{G}_{\mathrm{cl}}^{\mathrm{Tr}}(u_g)$=[(1,3),(2,2)] | $x_{5,6,9}$ |
| 10 | $^*x_8$=[(3,0,2), (3,0,2), $u_g$] | $\mathrm{G}_{\mathrm{cl}}^{\mathrm{Tr}}(u_g)$=[(0,2)] | $x_{5,6,8}$ |
| 11 | empty | | $x_{5,6,8}$ |

Table 1: Trace of $Open$ and $Sols$ in each iteration (It.) of NWMOA*. We mark extracted node of each iteration with symbol $^*$. The third column shows changes on $\mathrm{G}_{\mathrm{cl}}^{\mathrm{Tr}}$ lists.

NWMOA* by solving a sample MOSP instance with three cost components, depicted in Figure 1. $u_s$ and $u_g$ denote *start* and *goal*, respectively. Although the graph contains negative cycle, the problem is bounded because the cycle is not on any *start-goal* path. For the states reachable from *start*, the triple inside the state denotes **h**-value, calculated in the higher level of NWMOA*. We briefly explain all iterations (It.) of NWMOA* for the given instance, with the trace of generated nodes and sets illustrated in Table 1. Since none of the nodes in this simple instance is quickly dominated, we do not display the changes on $\mathbf{g}_{\mathrm{last}}^{\mathrm{Tr}}$ of states.

It.1-4: The first four iterations of the algorithm expand all states on the $cost_1$-optimal path, namely $u_s$, $u_1$, $u_2$ and $u_g$. At the end of the fourth iteration, we have one truncated cost vector stored in the $\mathrm{G}_{\mathrm{cl}}^{\mathrm{Tr}}$ list of each of the expanded states, as shown in third column of Table 1. $x_5$ is our first solution.

It.5: $x_2$, the second descendant node of $x_0$, is extracted. $x_2$'s truncated cost $\mathrm{Tr}(\mathbf{g}(x_2))$ = (1,1) dominates the only vector available in $\mathrm{G}_{\mathrm{cl}}^{\mathrm{Tr}}(u_2)$. Thus, (1,1) replaces (2,4) in $\mathrm{G}_{\mathrm{cl}}^{\mathrm{Tr}}(u_2)$. $x_2$ is non-dominated and should be expanded. The new node $x_6$ is added to $Open$ upon expansion of $x_2$.

It.6: $x_6$ is extracted from $Open$. This node appears non-dominated, and is a tentative solution. $\mathrm{G}_{\mathrm{cl}}^{\mathrm{Tr}}(u_g)$ is then updated with the new truncated cost (2,2). The existing vector (3,5) is dominated and removed. $x_6$ is added to $Sols$.

It.7: NWMOA* undergoes the first expansion with state $u_3$ via $x_3$. Two new nodes, $x_7$ and $x_8$, are added to the queue.

It.8: Both $x_7$ and $x_8$ exhibit the same $f_1$-value. In the absence of tie-breaking, we may extract $x_7$ first. $x_7$ appears non-dominated. Thus, we can add its truncated cost (0,2) to $\mathrm{G}_{\mathrm{cl}}^{\mathrm{Tr}}(u_2)$. Its expansion leads to $u_g$ via the new node $x_9$.

It.9: Both $x_8$ and $x_9$ exhibit the same $f_1$-value. Assume $x_9$ is extracted first. $x_9$ is a non-dominated solution node, so we can update both $\mathrm{G}_{\mathrm{cl}}^{\mathrm{Tr}}(u_g)$ and $Sols$ with the new node. This solution does not dominate any previous solutions.

It.10: $x_8$ is the only node in the queue. $x_8$ is a non-dominated solution node, and its truncated cost (0,2) dominates all vectors in $\mathrm{G}_{\mathrm{cl}}^{\mathrm{Tr}}(u_g)$. $x_8$ should be added to $Sols$. However, the new solution $x_8$ dominates the previous solution $x_9$ (both showing the same $f_1$-value). Thus, $x_9$ is removed from $Sols$.

It.11: $Open$ is empty, and all optimal solutions are in $Sols$.

The example above shows how NWMOA* processes nodes in the order of their $f_1$-value. As we observed in It.9,

NWMOA* may perform extra expansions due to not breaking ties between nodes before extractions. More accurately, $x_7$ would not be expanded if we processed nodes lexicographically. However, we observed in It.10 how NWMOA* refines the optimal solution set in such circumstances.

## Always Logarithmic Dominance Check with $k = 3$

In case of three objectives, truncated cost vectors consist of two cost components only. Since $\mathrm{G}_{\mathrm{cl}}^{\mathrm{Tr}}$ lists store non-dominated truncated vectors in lexicographical order, we can achieve logarithmic-time dominance check via binary search. Let $\mathbf{v} = (g_2, g_3)$ be the truncated cost vector of $x$, and $\mathbf{v}' = (g_2', g_3')$ be the predecessor of $\mathbf{v}$ in $\mathrm{G}_{\mathrm{cl}}^{\mathrm{Tr}}(s(x))$, obtained by binary search. Since $\mathbf{v}' \leq_{lex} \mathbf{v}$, we should have:

1. $g_2' = g_2$ and $g_3' = g_3$: thus $\mathbf{v}'$ weakly dominates $\mathbf{v}$
2. or $g_2' = g_2$ and $g_3' < g_3$: thus $\mathbf{v}'$ dominates $\mathbf{v}$
3. or $g_2' < g_2$: there are two cases, either $g_3' \leq g_3$ or $g_3' > g_3$. The former denotes $\mathbf{v}'$ dominates $\mathbf{v}$. The latter, however, confirms $\mathbf{v}$ is non-dominated, because all other lexicographically smaller candidates in $\mathrm{G}_{\mathrm{cl}}^{\mathrm{Tr}}(s(x))$, i.e., vectors before the predecessor $\mathbf{v}'$, do not exhibit smaller $cost_3$ than $g_3'$ and thus cannot dominate $\mathbf{v}$.

## Theoretical Results

This section provides a formal proof for the correctness of multi-objective search of NWMOA*, providing theoretical

results on why NWMOA* can solve bounded MOSP instances with negative weights. That is, we assume negative cycles have already been handled via Algorithm 1.

**Lemma 1** Suppose NWMOA*'s search is led by smallest (possibly negative) $f_1$-values. Let $x_i$ and $x_{i+1}$ be nodes extracted from $Open$ in two consecutive iterations of NWMOA*. We have $f_1(x_i) \leq f_1(x_{i+1})$ if $h_1$ is consistent.
**Proof Sketch** We distinguish two cases: i) if $x_{i+1}$ was available in $Open$ at the time $x_i$ was extracted, the lemma is trivially true. ii) otherwise, $x_{i+1}$ is the descendant node of $x_i$. For the edge linking state $s(x_i)$ to its successor $s(x_{i+1})$, the consistency requirement of $h_1$ ensures $h_1(s(x_i)) \leq h_1(s(x_{i+1})) + cost_1(s(x_i), s(x_{i+1}))$. Adding the cost $g_1(x_i)$ to both sides of the inequality yields $f_1(x_i) \leq f_1(x_{i+1})$. $\square$
**Corollary 1** Let $(x_1, x_2, ..., x_t)$ be the sequence of nodes extracted from $Open$. The (perfect) heuristic function **h** is consistent and admissible. Then, under the premises of Lemma 1, $i \leq j$ implies $f_1(x_i) \leq f_1(x_j)$, meaning $f_1$-value of extracted nodes are monotonically non-decreasing.

**Lemma 2** Suppose $x_j$ is extracted after $x_i$ and $s(x_i) = s(x_j)$. $x_i$ (weakly) dominates $x_j$ if $\text{Tr}(\mathbf{g}(x_i)) \preceq \text{Tr}(\mathbf{g}(x_j))$.
**Proof Sketch** $x_j$ is extracted after $x_i$, so we have $f_1(x_i) \leq f_1(x_j)$ according to Corollary 1. Given $h_1(s(x_i)) = h_1(s(x_j))$, we obtain $g_1(x_i) \leq g_1(x_j)$. The other condition $\text{Tr}(\mathbf{g}(x_i)) \preceq \text{Tr}(\mathbf{g}(x_j))$ means $g_2(x_i) \leq g_2(x_j), \ldots, g_k(x_i) \leq g_k(x_j)$. Thus, $\mathbf{g}(x_j)$ is no smaller than $\mathbf{g}(x_i)$ in all dimensions. $\square$

**Lemma 3** Suppose $x_j$ is extracted after $x_i$ with $s(x_i) = goal$. Extending $x_j$ towards the $goal$ state will not lead to a non-dominated solution node if $\text{Tr}(\mathbf{g}(x_i)) \preceq \text{Tr}(\mathbf{f}(x_j))$.
**Proof Sketch** Node $x_j$ is extracted after $x_i$, so we have $f_1(x_i) \leq f_1(x_j)$, or equivalently $g_1(x_i) \leq f_1(x_j)$. Expanding the condition $\text{Tr}(\mathbf{g}(x_i)) \preceq \text{Tr}(\mathbf{f}(x_j))$, we have $g_2(x_i) \leq f_2(x_j), \ldots, g_k(x_i) \leq f_k(x_j)$. Thus, $\mathbf{f}(x_j)$ is no smaller than $\mathbf{g}(x_i)$ in all dimensions. Since **h** is admissible, we can guarantee that all subsequent expansions of $x_j$ towards $goal$ will exhibit the same condition. Thus, $x_j$ and all its descendant nodes will be (weakly) dominated by $x_i$. $\square$

**Lemma 4** Dominated nodes cannot lead to any **cost**-optimal $start$-$goal$ solution path.
**Proof Sketch** We prove this lemma by assuming the contrary, namely by claiming that dominated nodes can lead to a **cost**-optimal $start$-$goal$ solution paths. Let $x$ and $y$ be two nodes associated with the same state where $y$ is dominated by $x$. Suppose that $\pi^*$ is a **cost**-optimal $start$-$goal$ solution path via the dominated node $y$. Since $x$ dominates $y$, one can replace the subpath represented by $y$ with that of $x$ on $\pi^*$ to further reduce the cost of the $start$-$goal$ optimum path for at least one attribute. However, being able to reduce the cost of the established optimal solution path would contradict our assumption on the optimality of the solution path $\pi^*$. Therefore, we conclude that dominated nodes cannot form any **cost**-optimal $start$-$goal$ solution path. $\square$

**Lemma 5** Let $y$ be a node weakly dominated by node $x$ and $s(x) = s(y)$. If $y$'s expansion leads to a **cost**-optimal solution path, $x$'s expansion will also lead to an optimal solution.
**Proof Sketch** We prove this lemma by assuming the contrary, namely that $x$ cannot lead to any **cost**-optimal solution path. Since $y$ is weakly dominated by $x$, we have $g_1(x) \leq g_1(y), \ldots, g_k(x) \leq g_k(y)$, meaning that $x$ offers a better cost at least in one dimension, or **cost** equal to the **cost** of $y$. In such condition, one can replace the partial path represented by $y$ with that of $x$, and nominate a path lexicographically smaller than or equal to the optimal solution path via $y$. The existence of a better solution path in the former case would contradict our assumption on the optimality of the solution path via node $y$. The later case means both paths are equal in terms of **cost** and thus should be non-optimal, contradicting our assumption. Therefore, node $x$ will definitely lead to a **cost**-optimal solution path if node $y$ leads to an optimal solution. $\square$

**Theorem 1** NWMOA* computes a **cost**-unique Pareto-optimal solution set for any bounded MOSP instance.
**Proof Sketch** NWMOA* enumerates all partial paths from the $start$ state towards the $goal$ state in best-first order, in search of all optimal solutions. The dominance rules utilised by NWMOA* (Lemmas 2 and 3) ensure that removal of (weakly) dominated nodes is safe, as they will not lead to **cost**-unique optimal solution paths (Lemmas 4 and 5). Thus, we just need to show that $Sols$ does not contain a (weakly) dominated solution when NWMOA* terminates. NWMOA* captures all nodes reaching the $goal$ state. However, since it does not process nodes lexicography, some tentative solutions may later appear dominated. Let $x$ be a new non-dominated solution extracted after solution $z$. We have $f_1(z) \leq f_1(x)$. There are two cases: i) if $f_1(z) < f_1(x)$, the tentative solution $x$ confirms the optimality of $z$ because the older solution $z$ cannot be dominated by $x$ and all future solutions. ii) if $f_1(z) = f_1(x)$, a dominance check is performed to ensure the optimality of $z$, or remove $z$ from $Sols$ if it is deemed to be (weakly) dominated by $x$, as scripted in lines 15-20 of Algorithm 2. Therefore, we conclude that NWMOA* terminates with returning a set of **cost**-unique Pareto-optimal solution set, even with negative weights. $\square$

## Experimental Results

This section compares the performance of NWMOA* against the recent MOSP algorithms: LTMOA* (Hernández et al. 2023), TMDA (Maristany de las Casas et al. 2023), EMOA* (Ren et al. 2022) and also the lazy variant of NAMOA*$_{dr}$ (Pulido, Mandow, and Pérez-de-la-Cruz 2015b) studied in Maristany de las Casas et al. (2023).

**Implementation:** We implemented our NWMOA* algorithm in C++ and used the publicly available version of the other algorithms, all implemented in C++. The implementation of TMDA and Lazy-NAMOA*$_{dr}$ utilises linked lists to store truncated vectors. However, we found both algorithms performing faster when vectors are handled via dynamic arrays (nearly 30%). Hence, we used the faster array-based implementation. For the LTMOA* algorithm, we were unable to obtain the original implementation and instances. We, therefore, implemented the fast performing variant of the algorithm (Lazy-LTMOA*-A) in C++ based on the descriptions provided in the original paper. All algorithms utilise dynamic arrays to store truncated vectors. We ran all experiments on a single core of an Intel Xeon Gold 5220R processor running at 2.2 GHz and with 32 GB of RAM, under the

CentOs Linux 7 environment and with a two-hour timeout. All C++ code was compiled using the GCC7.5 compiler.

**The _Open_ list:** Since NWMOA* processes nodes based on their $f_1$-value only, we can utilise simpler yet efficient data structures to implement the priority queue, such as bucket-based queues (Denardo and Fox 1979; Cherkassky, Goldberg, and Radzik 1996). It can be shown that the difference between the largest and smallest $f_1$-values available in _Open_ in any arbitrary iteration of NWMOA* is bounded by $max\{h_1(v) - h_1(u) + cost_1(u,v)|(u,v) \in E, h_1(v) \neq \infty\}$. Thus, we implemented _Open_ using a cyclic fixed-size bucket-based queue, with bucket width of one. Linked lists were used to handle queue operations via the Last-In, First-Out (LIFO) strategy.

**Benchmark instances:** We used the New York map from the 9th DIMACS Implementation Challenge: Shortest Paths[1] to generate MOSP instances. The map contains two cost components only: distance and time. To extend the dimensions, following Storandt (2012), we enriched the map with Shuttle Radar Topography Mission[2] height information and set the third dimension to be the positive height difference of the endpoints of each link, i.e., we set $cost_3(u,v) = |height(v) - height(u)|$ for each $(u,v) \in E$. For the fourth edge attribute, following Ren et al. (2022), we calculate the average (out)degree of the link, i.e., number of adjacent vertices of each end point, namely $cost_4 = \lfloor(deg(u) + deg(v))/2\rfloor$ for each $(u,v) \in E$. Finally, as in Maristany de las Casas et al. (2023), we set the fifth cost of each edge to 1, with $cost_5$ of paths denoting the number of edges traversed. We then generated 100 random $(start, goal)$ pairs, and evaluated all algorithms on the same set of instances but with 3-5 cost components. Our code and benchmark instances will be made publicly available.

Since all algorithms use the same approach to compute heuristic functions, we report the runtime of the main search only. Table 2 presents the runtime statistics for all of the studied algorithms with 3-5 cost components, as well as for the variant of NWMOA* with logarithmic-time dominance check for $k = 3$, denoted by NWMOA*$_{log}$. We report both arithmetic and geometric mean, and the runtime of unsolved cases is considered to be the timeout. $\phi$ in the last column represents the average slow-down factor (of runtime) over the mutually solved instances when compared to the virtual best oracle. The virtual oracle is given the best runtime of all algorithms for every mutually solved instance. $\phi = 1$ means the algorithm is as good as the virtual best oracle.

Comparing the performance of the algorithms with $k = 3$, we find that NWMOA*$_{log}$ is best performer, dominating other algorithms in all individual instances with $\phi = 1$. This variant performs on average 20% faster than the standard NWMOA*, 4.2 times faster than Lazy-LTMOA*, and above one order of magnitude faster than others. Our detailed results show that the number of dominance checks is reduced by 71% on average when binary search is used for dominance check in NWMOA* with $k = 3$. Comparing the results for instances with four and five cost compo-

[1]http://www.diag.uniroma1.it/ challenge9/download.shtml
[2]https://www2.jpl.nasa.gov/srtm/

| Algorithm | $|S|$ | Runtime(s) | | | | $\phi$ |
| | | Min. | Mean$_A$ | Mean$_G$ | Max. | |
|---|---|---|---|---|---|---|
| NY with 3 cost components (avg($|Sols|$)=5,090) | | | | | | |
| NWMOA*$_{log}$ | 100 | **0.01** | **3.2** | **1.1** | **15.1** | 1.0 |
| NWMOA* | 100 | 0.01 | 4.0 | 1.4 | 22.4 | 1.2 |
| L-LTMOA* | 100 | 0.05 | 14.1 | 4.7 | 75.5 | 4.2 |
| TMDA | 100 | 0.40 | 100.6 | 33.5 | 737.1 | 30.4 |
| L-NAMOA*$_{dr}$ | 100 | 0.37 | 109.7 | 36.2 | 845.5 | 33.0 |
| EMOA* | 100 | 2.59 | 671.1 | 225.5 | 4980.4 | 202.3 |
| NY with 4 cost components (avg($|Sols|$)=86,134) | | | | | | |
| NWMOA* | 100 | **0.17** | **728.0** | **146.0** | **5190.7** | 1.0 |
| L-LTMOA* | 97 | 0.39 | 1578.7 | 337.3 | 7200.0 | 2.3 |
| TMDA | 49 | 5.21 | 4779.0 | 2494.6 | 7200.0 | 37.8 |
| L-NAMOA*$_{dr}$ | 45 | 5.12 | 4860.3 | 2598.5 | 7200.0 | 41.3 |
| EMOA* | 27 | 23.35 | 5870.2 | 4276.7 | 7200.0 | 172.2 |
| NY with 5 cost components (avg($|Sols|$)=120,011) | | | | | | |
| NWMOA* | 82 | **0.21** | **2363.2** | **502.0** | 7200.0 | 1.0 |
| L-LTMOA* | 68 | 0.49 | 3125.6 | 891.4 | 7200.0 | 2.2 |
| TMDA | 33 | 7.09 | 5597.8 | 3631.8 | 7200.0 | 34.4 |
| L-NAMOA*$_{dr}$ | 31 | 7.45 | 5649.8 | 3738.5 | 7200.0 | 37.9 |
| EMOA* | 17 | 31.31 | 6301.7 | 5179.3 | 7200.0 | 162.3 |

Table 2: Runtime statistics of the algorithms (in seconds) with $k$=3,4,5. We report both Arithmetic (Mean$_A$) and Geometric (Mean$_G$) Mean. $|S|$ is the number of solved cases (out of 100), and $\phi$ shows the average slowdown factor of mutually solved cases compared to the virtual best oracle. The runtime of unsolved cases is considered to be two hour.

nents, we can find NWMOA* performing better in every mutually solved instances with $\phi = 1$. Extending dimensions of the problem would not only enlarge the search space, but also make dominance checks more expensive. Nonetheless, NWMOA* consistently outperforms other algorithms for $k = 4, 5$. It is more than two times faster than Lazy-LTMOA*, and above one order of magnitude faster than others. Note that our comparison with Lazy-LTMOA* can be considered head-to-head since it is implemented on the same framework as NWMOA*.

**Ablation study:** To gain a deeper understanding of how the key features of NWMOA* impact search performance, we performed detailed performance analyses on three variants of NWMOA*, namely: NWMOA* without quick dominance check and/or NWMOA* without lexicographical ordering of node in $G_{cl}^{Tr}$. We also compare how Lazy-LTMOA* performs against these variants. The detailed results are presented in Table 3 for mutually solved instances for $k = 4$, where $\phi$ denotes average percentage increase in runtime when compared to standard NWMOA*. $|per|$ also denotes the average number of percolations (total swaps performed) in the binary heap of Lazy-LTMOA*, and the total number of buckets traversed in the priority queue of NWMOA*.

Comparing the results, we see all variants expanding the same number of nodes on average (slightly larger than Lazy-LTMOA*), and consuming nearly the same amount of memory. Interestingly, the quick dominance check of NWMOA* effectively prunes two third of dominated nodes, reducing the number of generated nodes by 20% on average. We see

| Variant | $\phi$ | Mem. | $|generated|$ | $|expansions|$ | $|pruned_l|$ | $|pruned_q|$ | $|check|$ | $|per|$ |
|---|---|---|---|---|---|---|---|---|
| NWMOA* | - | 0.74 | $147.45 \times 10^6$ | $59.80 \times 10^6$ | $82.09 \times 10^6$ | $46.37 \times 10^6$ | $230.05 \times 10^9$ | $163.73 \times 10^3$ |
| NWMOA*$_{\text{w/o qdc}}$ | 18.81% | 0.70 | $185.33 \times 10^6$ | $59.80 \times 10^6$ | $125.45 \times 10^6$ | $0.00 \times 10^0$ | $256.28 \times 10^9$ | $163.73 \times 10^3$ |
| NWMOA*$_{\text{w/o lex}}$ | 107.81% | 0.74 | $147.45 \times 10^6$ | $59.80 \times 10^6$ | $82.09 \times 10^6$ | $43.37 \times 10^6$ | $302.81 \times 10^9$ | $163.73 \times 10^3$ |
| NWMOA*$_{\text{w/o (lex+qdc)}}$ | 108.83% | 0.70 | $185.33 \times 10^6$ | $59.80 \times 10^6$ | $125.45 \times 10^6$ | $0.00 \times 10^0$ | $310.85 \times 10^9$ | $163.73 \times 10^3$ |
| L-LTMOA* | 165.63% | 0.77 | $184.92 \times 10^6$ | $59.68 \times 10^6$ | $125.16 \times 10^6$ | $0.00 \times 10^0$ | $310.47 \times 10^9$ | $5.04 \times 10^9$ |

Table 3: NWMOA*'s performance compared against variants without quick dominance check (w/o qdc), and/or without lexicographical ordering (w/o lex), and L-LTMOA*. The results are for 98 mutually solved instances with $k = 4$. We report the averages of: percentage increase in runtime w.r.t. NWMOA* ($\phi$), memory (in GB), number of generated, expanded, linearly pruned ($pruned_l$), and quickly pruned ($pruned_q$) nodes, total dominance checks ($|check|$) and queue percolation ($|per|$).

Table 4: NWMOA*'s performance with three different types of priority queues, with/without tie breaking. Results for 71 mutually solved instances of the randomised NY map with 4 cost components. We show averages of runtime (in seconds), memory (in GB), number of expansions ($|exp|$), and queue percolation ($|per|$).

| Queue | Runtime | Mem. | $|exp|$ | $|per|$ |
|---|---|---|---|---|
| Bucket-LIFO | 997.3 | 1.3 | $123.9 \times 10^6$ | $2.7 \times 10^3$ |
| Bucket-FIFO | 1270.6 | 1.7 | $135.0 \times 10^6$ | $2.7 \times 10^3$ |
| Hybrid w/o tie | 1078.9 | 1.4 | $123.9 \times 10^6$ | $0.2 \times 10^9$ |
| Hybrid w tie | 1343.0 | 2.8 | $122.5 \times 10^6$ | $5.2 \times 10^9$ |
| Heap w/o tie | 1434.5 | 1.5 | $127.1 \times 10^6$ | $7.2 \times 10^9$ |
| Heap w tie | 1424.8 | 2.8 | $122.5 \times 10^6$ | $6.8 \times 10^9$ |

18.8% increase in runtime when no quick dominance check is in place. The impact is more severe when non-dominated truncated costs are not stored lexicographically, where we observe a runtime increase of over 100%. In this situation, the number of dominance checks is raised drastically, requiring a linear traversal of the entire $G_{cl}^{Tr}$ list. The performance decline is less pronounced if quick dominance checks are not performed alongside the latter case (the second last variant of Table 3). Even with both features removed, this downgraded variant still preforms better than Lazy-LTMOA* due to its simpler node queueing strategy and far lower $|per|$.

**Negative weights and priority queues:** For the last experiment of this study, we analyse the performance of NWMOA* using different data structures for $Open$. Further, to evaluate our algorithm on large graphs with both negative and non-correlated edge attributes, we changed the edge cost vector of the NY map with four new attributes as follows: $cost_1$ is the energy consumption along the link. We use the energy model of Ahmadi et al. (2021b) to produce realistic, potentially negative, energy estimates for an electric vehicle with three passengers on board. We did not bound the battery capacity, thus the plain NWMOA* can be used to find energy efficient paths. The second cost is now a penalised height function. We set $cost_2$ of link $(u, v) \in E$ to be $height(v) - height(u)$ if $height(v) \leq height(u)$ (downhill links), and $2 \times (height(v) - height(u))$ otherwise (uphill links). For the third and fourth costs, we choose random integers in the [1,100] range. The resulting graph is free of negative cycles, thus all MOSP instances would be bounded.

We report in Table 4 the performance results for three types of priority queues in two variations: Bucket (with LIFO or First-In, First-Out strategy), Hybrid and Heap queues (with or without tie-breaking). Our Hybrid queue is a two-level bucket-based queue, with a bucket list and heap used for the higher and lower levels, respectively (Denardo and Fox 1979). Nodes may be inserted to either of the data structures (depending on $f_1$-value), but are always extracted from the lower level. We set the bucket width to one in both Bucket and Hybrid queues. Hybrid queue can handle both integer and non-integer costs. We use the same timeout and ($start$, $goal$) pairs as in the previous experiment, and report the average number of expansions, memory consumed, and queue percolation for all mutually solved instances.

As we expected, the minimum number of expansions is achieved when tie-breaking is in place for Heap and Hybrid queues. However, this seems costly in terms of space, with the memory consumption nearly doubled when compared with the variant without tie-breaking. Although we see slight improvement in the runtime of heap queue with tie-breaking, the effort is not paid off in the Hybrid queue and NWMOA* performs around 20% faster when not breaking ties. NWMOA* performs best with Bucket queue using the LIFO strategy. It undergoes minimal queue effort and memory use, while expanding only 1% extra nodes. The Bucket queue with FIFO strategy, however, shows the largest number of expansions, and is outperformed by Hybrid queue without tie-breaking. This observation highlights the significance of prioritising most recent (better informed) expansions in multi-objective search with A*.

## Conclusion

We have introduced NWMOA*, an exact MOSP algorithm that determines within polynomial time whether a point-to-point MOSP problem instance is bounded. If the instance is found to be bounded, NWMOA* can then compute a cost-unique Pareto-optimal solution set, even in the presence of negative weights. NWMOA* challenges the convention in multi-criteria search by not processing paths in lexicographical order of their costs, while utilising novel strategies to expedite the exhaustive search of A*. The results of our extensive experiments over a new large set of realistic instances show the success of NWMOA* in efficiently solving difficult MOSPP instances in limited time, outperforming all state-of-the-art algorithms by up to an order of magnitude.

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
