# OpenReview forum: "Exact Multi-objective Path Finding with Negative Weights and Negative  Cycles"
_icaps-conference.org/ICAPS/2024/Conference — ICAPS 2024_

### Official Review · Reviewer_o3ro · 2024-01-16

**Significance And Importance:** 2
**Soundness:** 3
**Novelty:** 2
**Clarity:** 3
**Overall Evaluation:** 1
**Confidence:** 5

**Weaknesses:**

0: Minor weaknesses requiring some work to be addressed for the paper to be accepted.

**Contributions Of The Paper:**

The paper describes a multi-objective search algorithm that can deal with negative edge weights, NWMOA. The algorithm can solve a problem instance if it is bounded, i.e., there is no negative cycle in any solution. Additionally, the paper describes several speed-up techniques for multi-objective search, including a quick dominance check approach and a bucket-based heap.

**Ethical Considerations:**

(1) Not Applicable: The paper does not have any ethical considerations to address

**Nomination For Best Paper:**

No

**Questions For Authors:**

Please address my point 4 in the weakness section. How are algorithms 1 and 4 actually implemented in the code?

**Reproducibility:**

4: Authors promise to release code and domains (whichever apply).

**Strengths Of The Paper:**

1. Almost all existing work on multi-objective search only considers non-negative weights, which limits their application. I believe that the contribution of this paper is interesting to the ICPAS audiences.
2. The paper is easy to follow and written well.
3. The experimental study is quite comprehensive. The ablation study was helpful for understanding where the improvement come from.

**Weaknesses Of The Paper:**

1. Although the paper is presented as a new MO search algorithm NWMOA that can handle negative weights, it seems more like a paper that describes several independent techniques. For example, the high level of NWMOA* can be combined with existing MO search algorithms like EMOA* and LTMOA* as well. The speed-up techniques also seem applicable to other MO search algorithms and are not related to the presence of negative weights.

2 Some of the proposed techniques have appeared in existing works. Although some of these papers are already cited, I think the authors should discuss them in more detail.
2.i) Using bucket queue and ordering nodes according to only f1 value has appeared in [1]. Although [1] focuses on a different problem, the Weight Constraint Shortest Path (WCSP) problem, it is not surprising that the technique is transferable given the close relation between WCSP and MOSP, as discussed in [1].
2.ii) EMOA*[2] also employed a log time dominance checks mechanism for problems with 3 objectives and skips checking those truncated cost vectors if their first components are larger than that of the vector being checked. Although EMOA* uses a different data structure to implement this idea, I think it should be mentioned in this paper.

3. It was not initially clear to me why consistent heuristics are always available for bounded problem instances, and I think the paper would benefit from adding such a discussion to the theoretical results.

4. Part of the pseudo-code is inconsistent with the text and seems inefficient.
4.i) On Lines 4-7 of algorithm 1, iterating over the entire S and computing a cost-i-optimal path seems extremely inefficient. However, what is described in the text (k runs of Bellman-Ford or Dijkstra’s) seems more reasonable.
4.ii) The remove and insert operations on Lines 7 and 9 of algorithm 4 seem to take O(N) time, given that dynamic arrays are used to implement V. Is NWMOA really implemented this way?

In the abstract and the conclusion, the improvement of performance seems to be over-claimed. For example, the abstract claims that NWMOA "outperforming the state of the art by up to an order of magnitude." If we look at the results, LTMOA seems to be the state-of-the art and is outperformed by at most 4 times.

[1] Ahmadi, Saman, et al. "Enhanced Methods for the Weight Constrained Shortest Path Problem: Constrained Path Finding Meets Bi-objective Search." arXiv preprint arXiv:2207.14744 (2022).
[2] Ren, Zhongqiang, et al. "Enhanced multi-objective A* using balanced binary search trees." Proceedings of the international symposium on combinatorial search. Vol. 15. No. 1. 2022.

---

> ### Author Rebuttal · Authors · 2024-01-26
>
> Thank you for the thoughtful comments and suggestions. We will provide additional explanations to address each concern, including the applicability of the proposed techniques to existing best-first multi-objective search approaches. We will also provide further elaboration on why consistent heuristics are always available in negative-cycle free graphs.
>
> Response to concern4a:
> It is indeed inefficient to calculate lower bounds for each state individually. Instead, for each objective, NWMOA* can utilise a modified version of the conventional one-to-all backward Dijkstra's algorithm, while allowing for re-expansions. During expansions, it examines the length (number of edges) of the extended path for a typical successor state 'u' to identify the presence of a negative cycle. If the length of the extended path grows to |S|, the backward search can be terminated as a negative cycle has been detected along a start-goal path (via u). Line 6 in Algorithm 1 corresponds to this scenario, where the algorithm can exit as soon as a negative cycle is observed during the backward one-to-all search. Therefore, lines 4-7 are implemented using a single-objective backward Dijkstra's algorithm, with line 6 serving as the exit condition for detecting the first negative cycle. We will provide a more clearer explanation of this in the description of Algorithm 1.
>
> Response to concern4b:
> Correct. These operations may result in linear time complexity as items are moved forward or backward in the dynamic array. However, as indicated in Table3, this additional cost is justified by the efficiency gained from performing lower dominance checks in multi-objective search, thanks to the lexicographic ordering of truncated cost vectors. The NWMOA* (w/o lex) variant (in Table3) in fact adopts the ordering approach proposed for LTMOA*, where insertions and removals always occur at the end of the list. According to the results, NWMOA* demonstrates improved performance when cost vectors are maintained in lexicographic order via dynamic arrays.
>
> Response to: "the improvement of performance"
>
> NWMOA* performs an order of magnitude faster than TMDA and up to 4 times better than LTMOA*, two state-of-the-art MOSP algorithms.
> We will make the performance improvement clearer in both abstract and conclusion.

---

### Official Review · Reviewer_UjZZ · 2024-01-23

**Significance And Importance:** 2
**Soundness:** 3
**Novelty:** 3
**Clarity:** 4
**Overall Evaluation:** 2
**Confidence:** 3

**Weaknesses:**

0: Minor weaknesses requiring some work to be addressed for the paper to be accepted.

**Contributions Of The Paper:**

This paper introduces NWMOA*, an algorithm designed to address the point-to-point Multi-objective Shortest Path (MOSP) Problem in scenarios where the graph permits negative weights and cycles. A key contribution lies in presenting an algorithm for detecting negative cycles (Algorithm 1). Subsequently, the paper rigorously establishes the correctness of NWMOA* (Theorem 1, after several lemmas obtained), asserting that the algorithm computes a cost-unique Pareto-optimal solution set for any bounded MOSP instance, considering the detection and handling of negative cycles having already been done. Furthermore, the paper incorporates various techniques within the A* search process to expedite the algorithm. Extensive experiments are conducted, yielding promising results.

**Ethical Considerations:**

(1) Not Applicable: The paper does not have any ethical considerations to address

**Nomination For Best Paper:**

No

**Questions For Authors:**

1) The title of the paper highlights the importance and novelty of "negative cycles," but based on Algorithm 1, isn't it *almost* trivial to detect them?

2) If "negative weight" has not been considered before, how is it possible that the current algorithm is compared with other existing algorithms that, by nature, do not take negative weights into account?

**Reproducibility:**

4: Authors promise to release code and domains (whichever apply).

**Strengths Of The Paper:**

The paper is commendably structured and presented, offering a detailed exposition of the proposed algorithm and several enhancements aimed at improving efficiency.

The algorithm is accompanied by a theoretical proof, bolstering confidence in its correctness.

While the reviewer did not delve into the proof's details, there's a sense of assurance regarding the algorithm's correctness. Comparative experiments, along with a provided example, further support the paper's robustness.

**Weaknesses Of The Paper:**

A "potential" weakness is identified, acknowledging first that it might stem from a lack of detailed scrutiny or the reviewer's non-specialization in the field. Specifically, the significance of Algorithm 1, particularly the part detecting negative cycles, is questioned. It appears to be rather straightforward, or not?

Additionally, the reviewer expresses curiosity about the perceived importance of negative weights and suggests a potential solution involving the addition of a base value to all edges to eliminate negative costs without altering solution optimality. It is most-likely a naive question?

I understand that the actual challenge of negative weights might not be such a trivial matter. If it indeed presents a challenge, how is it possible that NWMOA* is compared with all the other existing algorithms (such as EMOA*, TMDA, and LTMOA*), while these algorithms are unable to handle negative weights at all?

---

> ### Author Rebuttal · Authors · 2024-01-26
>
> Thanks for the considerate comments and suggestions.
>
> Response to: "the addition of a base value to all edges to eliminate negative costs"
>
> This method does not always work for negative-weight graphs and may lead to incorrect solutions in some cases. Please see our response to Question#2 of Reviewer#1, where we discuss a scenario with an incorrect optimal solution produced by this method.
>
>
> Response to Q1:
> Negative cycles can render MOSP problems unbounded by providing cyclic optimum paths with at least one objective approaching negative infinity. However, the presence of negative cycles in a graph does not necessarily imply that all MOSP problems are unbounded.
> As explained in the paper, negative cycles are not problematic as long as they do not appear on any start-goal paths. Therefore, the objective in Algorithm 1 is not solely to detect any negative cycles, but specifically those that have the potential to make MOSP unbounded.
> As outlined in Algorithm 1, this task can be integrated into the procedure required by NWMOA* to construct its heuristic function, so we can terminate early, without conducting an exhaustive multi-objective search, if a negative cycle can be found on any start-goal paths.
>
> Response to Q2:
> NWMOA* is capable of handling MOSP instances with or without negative weights, making it suitable for comparison with existing solutions on MOSP instances with non-negative weights. In order to benchmark against recent solutions, our initial set of experiments focused on instances with non-negative weights only. This approach allowed us to create bounded MOSP problems, enabling a fair comparison of the multi-objective search effort (Algorithm 2) across different algorithms. Nevertheless, since NWMOA* is developed on the standard A* search framework, we believe that the techniques proposed in this paper have the potential to benefit existing algorithms, whether in handling negative weights or improving their overall search performance.

---

### Official Review · Reviewer_uodv · 2024-01-23

**Significance And Importance:** 2
**Soundness:** 3
**Novelty:** 3
**Clarity:** 3
**Overall Evaluation:** 1
**Confidence:** 3

**Weaknesses:**

0: Minor weaknesses requiring some work to be addressed for the paper to be accepted.

**Contributions Of The Paper:**

The paper left me somewhat confused. The paper is motivated by solving multi-objective path finding with negative weights. However, the proposed algorithm, according to the largest part of the experiments, seems to outperform state-of-the-art algorithms. Only a small part of the experiments evaluate the proposed algorithm on a graph with negative weights. Therefore, I would consider changing the title and focus of this paper. The paper is mostly well-written, but some information is missing for the paper to be self-contained. The theoretical part seems sound, and the experimental section is convincing.

**Ethical Considerations:**

(1) Not Applicable: The paper does not have any ethical considerations to address

**Nomination For Best Paper:**

No

**Questions For Authors:**

1) Does it matter which cost the algorithm chooses to be f_1? Is it better to choose a cost for f_1 that has many different values in the graph or one that has only a few different values? Did you try different policies for choosing this cost?
2) Assume we modify the given graph and add a large value to all edges, making them non-negative. In this case, will all state-of-the-art MOPS algorithms also be able to return all the Pareto Frontier solutions (assuming no negative cycles/the high-level is executed before)? In general, it seems that all state-of-the-art algorithms can be adjusted to solve multi-objective path finding with negative weights. What makes these algorithms (as they are) unable to find all solutions in this setting? Can you provide an example of where these algorithms don't return the same solution as MWMOA*?

--- Post-rebuttel ---
Thank you for answering my questions. I suggest adding a short discussion to the paper on the fact that adjusting existing MOSP algorithms to solve MOSP with negative cost is not straightforward.

**Reproducibility:**

4: Authors promise to release code and domains (whichever apply).

**Strengths Of The Paper:**

- The paper presents a novel algorithm that outperforms state-of-the-art algorithms.
- The new algorithm can solve problems with negative weights and cycles.
- The paper is well-written and well-organized in most parts.
- The experiments section is convincing.

**Weaknesses Of The Paper:**

Main comments -
- There is some information missing, which makes some parts hard to understand (all mentioned below).
- As I mentioned above, I find the motivation of the paper a bit confusing. While the paper suggests a method that can handle negative weights, most of the experimental study evaluates this method on graphs with no negative weights. Moreover, I couldn't understand whether all state-of-the-art MOSP algorithms are also capable of solving MOSP (with only minor modifications, where no negative cycles exist).

Other comments and suggestions -
- Introduction - bounded MOSP is mentioned but the reader cannot understand what it is.
- Introduction - "both Negative Weights and negative cycles" -> "both negative weights and negative cycles".
- Introduction - "realistic instances show the" - "realistic instances shows the".
- Introduction - "a pair of (Origin, Destination)" - in the rest of the paper, they are called start and goal.
- Notation and Problem Formulation - "which every individual solution offers a path that minimises the multi-criteria problem in all dimensions" - this should be explained. I couldn't understand what happens with negative cycles. I only understood it while reading the suggested algorithm itself.
- Notation and Problem Formulation - The operator Tr(v) is very hard to understand. It should be better defined.
- Notation and Problem Formulation - Definition hi - "cost_p" -> "cost_i". Also, the notation of cost_i() is overloaded. How does it give the cost from some state u to goal without receiving it?
- NWMOA*’s High-Level Description - This is basically a way to calculate the perfect heuristic and, during this calculation, it also finds negative cycles. The name "high-level" is misleading.
- Multi-objective Search of NWMOA* - "state and insert it into the priority" - "state and inserts it into the priority".
- Algorithm 2 - line 3 - g^Tr_last is a vector and, therefore, a value cannot be assigned to it.

---

> ### Author Rebuttal · Authors · 2024-01-26
>
> We are grateful for the thoughtful feedback provided on our paper. We will ensure all the comments and confusions are rectified in the revision.
>
> Respone to: "How does cost_i give the cost from some state u to goal without receiving it?"
>
> Algorithm1 ensures that all cost_i-values are received before we give them to NWMOA* as lower bounds (h_i-values).
>
> Response to Q1:
> Changing the order of objectives does impact the performance, but our preliminary experiments show that there is no direct relationship between the range of f_1-values and search performance. We did not use any form of heuristics to reorder the objectives in the experiments and followed the conventional objective ordering used in the literature for instances designed on the DIMACS maps.
>
> Response to Q2:
> Adapting existing solutions to MOSP with negative weight is not straightforward and cannot be accomplished by simply reweighting the edges of the graph. Technically, uplifting edges with a large value can result in incorrect solutions, even in the single-objective shortest path problem.
> Consider a simple triangular-like graph with three vertices and three edges all with cost=-1.
> Edges are {(start,vertex1), (vertex1,goal), (start,goal)}; forming two simple start-goal paths:
>
> path1:start->vertex1->goal;cost=-2
>
> path2:start->goal;cost=-1
>
> path1 here shows a lower cost and is optimal.
> One can shift all edges by one unit (or larger values) to get rid of negative weights (all edges would show cost=0). We then have:
>
> path1:start->vertex1->goal;cost=0
>
> path2:start->goal;cost=0
>
> which introduces both path1 and path2 as cost-optimal solutions, making the technique incorrect in handling negative weights by returning non-optimal paths as an optimal path.

---

### Meta-Review · Area_Chair_TdYR · 2024-02-05

**Recommendation:** Accept (Poster)
**Confidence:** 4

**Metareview:**

This paper introduces NWMOA*, an algorithm designed to address the point-to-point Multi-objective Shortest Path (MOSP) Problem in scenarios where the graph permits negative weights and cycles.  A key contribution lies in presenting an algorithm for detecting negative cycles (Algorithm 1). Subsequently, the paper rigorously establishes the correctness of NWMOA* (Theorem 1, after several lemmas obtained), asserting that the algorithm computes a cost-unique Pareto-optimal solution set for any bounded MOSP instance, considering the detection and handling of negative cycles having already been done. Furthermore, the paper incorporates various techniques within the A* search process to expedite the algorithm. Extensive experiments are conducted, yielding promising results.  Additionally, the paper describes several speed-up techniques for multi-objective search, including a quick dominance check approach and a bucket-based heap.

Strengths
- The paper presents a novel algorithm that outperforms state-of-the-art algorithms.
- The new algorithm can solve problems with negative weights and cycles.
- The paper is well-written and well-organized in most parts.
- The experiments section is convincing.
- The paper offers a detailed exposition of the proposed algorithm and several enhancements aimed at improving efficiency.
- The algorithm is accompanied by a theoretical proof, bolstering confidence in its correctness.


Weaknesses
- Some of the proposed techniques have appeared in existing works. Although some of these papers are already cited, we think the authors should discuss them in more detail.
- Using bucket queue and ordering nodes according to only f1 value has appeared in [1]. Although [1] focuses on a different problem, the Weight Constraint Shortest Path (WCSP) problem, it is not surprising that the technique is transferable given the close relation between WCSP and MOSP, as discussed in [1].
- EMOA*[2] also employed a log time dominance checks mechanism for problems with 3 objectives and skips checking those truncated cost vectors if their first components are larger than that of the vector being checked. Although EMOA* uses a different data structure to implement this idea, we think it should be mentioned in this paper.
[1] Ahmadi, Saman, et al. "Enhanced Methods for the Weight Constrained Shortest Path Problem: Constrained Path Finding Meets Bi-objective Search." arXiv preprint arXiv:2207.14744 (2022).
[2] Ren, Zhongqiang, et al. "Enhanced multi-objective A* using balanced binary search trees." Proceedings of the international symposium on combinatorial search. Vol. 15. No. 1. 2022.
- It was not initially clear to us why consistent heuristics are always available for bounded problem instances, and we think the paper would benefit from adding such a discussion to the theoretical results.
- In the abstract and the conclusion, the improvement of performance seems to be over-claimed. For example, the abstract claims that NWMOA "outperforming the state of the art by up to an order of magnitude." If we look at the results, LTMOA seems to be the state-of-the art and is outperformed by at most 4 times.

Suggestions to improve the final submission:
- We suggest adding a short discussion on why adjusting existing MOSP algorithms to solve MOSP with negative costs is not straightforward.
- Add a discussion regarding the existing techniques that share some similarities with the ones proposed in this paper, for instance [1] and [2] above.
- The authors stated in the rebuttal "NWMOA* performs an order of magnitude faster than TMDA and up to 4 times better than LTMOA*, two state-of-the-art MOSP algorithms. We will make the performance improvement clearer in both abstract and conclusion."
- The authors state in the rebuttal, "our initial set of experiments focused on instances with non-negative weights only". We would make this clearer so as not to confuse the reader. It would also be good to have a set of experiments that include negative weights.

The paper does not need an ethical statement to accompany it.

**Ethical Considerations:**

(1) Not Applicable: The paper does not have any ethical considerations to address